## Research Article

global mental health; caregiver; public health; low-income countries; interventions

**Corresponding author:**
Alethea Desrosiers;
Email: alethea_desrosiers@brown.edu

# Exploring potential mental health spillover effects among caregivers and partners of youth in Sierra Leone: A qualitative study

Alethea Desrosiers[1] , Carolyn Schafer[2], Laura Bond[3], Adeyinka Akinsulure-Smith[4], Miriam Hinton[5], Alpha Vandi[5] and Theresa S. Betancourt[3]

[1]Department of Psychiatry and Human Behavior, Brown University, Providence, RI, USA; [2]Institute for Public Health and Medicine, Northwestern University, Evanston, IL, USA; [3]Boston College School of Social Work, Chestnut Hill, MA, USA; [4]Department of Psychology, City College of New York, NY, USA and [5]Caritas Sierra Leone, Freetown, Sierra Leone

## Abstract

Given the large mental health treatment gap in low- and middle-income countries (LMICs), particularly in post-conflict settings like Sierra Leone, and the limited healthcare infrastructure, understanding the wider benefits of evidence-based mental health interventions within households is critical. This study explored potential mental health spillover effects – the phenomenon of beneficial effects among nonparticipants – among cohabitating caregivers and partners of youth who participated in an evidence-based mental health intervention in Sierra Leone. We recruited a sub-sample of cohabitating caregivers and partners (*N* = 20) of youth intervention participants; caregivers had enrolled in a larger study investigating indirect benefits of the evidence-based intervention in Sierra Leone (MH117359). Qualitative interviews were conducted at two time points to explore the following: (a) potential mental health spillover effects and (b) through which mechanisms spillover may have occurred. Two trained coders reviewed transcripts and analyzed qualitative data, assisted by MaxQDA. Qualitative findings suggested that spillover effects likely occurred and supported three potential mechanisms: decreased caregiving burden, behavior changes among Youth Readiness Intervention participants and improved interpersonal relationships. Mental health spillover effects may occur following youth intervention participation in a post-conflict LMIC. Investing in evidence-based services may offer indirect benefits that extend beyond those directly receiving services.

## Impact statement

The mental health treatment gap continues to be a major problem in low- and middle-income countries (LMICs), and individuals who need mental health services struggle to access formal supports. Investigating how participation in mental health services could have a wider societal benefit through indirect pathways (e.g., household spillover effects) could help strengthen the case among government stakeholders to invest in mental health infrastructure and strengthen policies supporting mental health service provision. Spillover effects are the phenomenon of beneficial effects of an intervention experienced by nonparticipants. This article presents a deeper dive into how participation in a mental health intervention might have much wider benefits for society beyond the benefits for individual participants. We conducted interviews with caregivers of youth (aged 18–30) in Sierra Leone who participated in an evidence-based intervention, the Youth Readiness Intervention, to explore whether improvements in youth behavior and functioning helped to alleviate a sense of caregiver burden. Prior research on spillover effects has primarily focused on health effects rather than mental health. Providing evidence for the phenomenon of mental health spillover effects in a rural region of Sierra Leone could help influence decisions among policy makers to increase investments in mental health services in LMICs and other low-resource settings globally.

## Exploring potential mental health spillover effects among caregivers and Partners of Youth in Sierra Leone: A qualitative study

Mental health disorders are one of the largest contributors to the global burden of disease among youth and adults (GBD 2019 Mental Disorders Collaborators, 2022). This burden is compounded in low- and middle-income countries (LMICs) and other low-resource settings due to the widening mental health treatment gap (Mathers and Loncar, 2006; World Health Organization, 2008). While there are evidence-based mental health interventions that have demonstrated feasibility and effectiveness in LMICs (Barry et al., 2013; Betancourt et al., 2014; Fazel et al., 2014), human and financial resource constraints can severely impede their reach and

sustainability. For example, in Sierra Leone, a small country in West Africa that has endured numerous hardships (i.e., civil war, the Ebola epidemic, and political instability), the mental health treatment gap is estimated at 98% (Alemu et al., 2012). Significant limitations in the nation's health infrastructure have made it difficult to effectively address the unmet mental health needs of the population (Alemu et al., 2012; Yoder et al., 2016; Hopwood et al., 2021). In Sierra Leone, these limitations are further compounded by negative beliefs associated with poor mental health and the stigma regarding mental illness (Akinsulure-Smith and Conteh, 2018). Gaining a better understanding of "whether" and "how" evidence-based mental health interventions might offer wider societal benefits and reach a larger segment of the population could help address resource constraints in LMICs and spur stakeholder investments in scaling out mental health interventions.

Accounting for spillover effects of evidence-based interventions across households and communities is one potential strategy to factor in wider societal benefits of mental health interventions. Spillover is the phenomenon of beneficial intervention effects to nonparticipants (Brouwer et al., 2009; Al-Janabi et al., 2016). Evidence-based mental health interventions delivered to youth may indirectly improve the mental health of other household members (Bobinac et al., 2010; Bobinac et al., 2011; Price et al., 2015). Research on spillover effects of mental health *interventions* is sparse (McBain et al., 2015; Al-Janabi et al., 2016), particularly in low-resource settings, where maximizing benefit and minimizing cost are essential (Lancet Global Mental Health Group et al., 2007; Chisholm et al., 2016). With the high costs associated with improving the health of people with mental health problems in LMICs (Kieling et al., 2011; Pedersen et al., 2019), measuring household spillover effects of mental health interventions could influence mental health policy decision-making and government investments in mental health service provision.

Studies on mental health spillover effects in high-income countries have found that family members living with individuals experiencing mental health problems report significantly poorer quality of life and health status, as well as poorer mental health (Henry and Cullinan, 2021; Lee et al., 2022). Considering that mental and/or behavioral issues in one family member can exacerbate the health and well-being of other household members, implementing/increasing access to evidence-based interventions to promote mental health and functioning among youth exhibiting mental health problems could "spillover" into the household and benefit primary caregivers and/or partners.

Preliminary quantitative data from one prior study in Sierra Leone indicate that caregivers of Sierra Leonean youth aged 15–24 who participated in an evidence-based mental health intervention – the Youth Readiness Intervention (YRI) – delivered within school settings (Betancourt et al., 2014) experienced reduced emotional distress and burden of care compared with caregivers of youth in the control condition (McBain et al., 2015). Investigating household spillover effects of the YRI are highly relevant because the intervention has demonstrated effectiveness in improving emotion regulation skills, prosocial functioning and functional impairment among youth who received the intervention in school settings compared with control youth (Betancourt et al., 2014). Additionally, in a hybrid-implementation effectiveness trial of the YRI integrated within entrepreneurship training for high-risk Sierra Leonean youth aged 18–30, those who participated in the YRI reported significant improvements in depression and anxiety symptoms compared with control youth (Freeman et al., under review, see Betancourt et al., 2021 for study protocol).

Based on our review of the literature, there are no other studies specifically on the spillover effects of *mental health* interventions in LMICs. While this prior study provides some initial support for the occurrence of mental health spillover effects within households in Sierra Leone, the specific mechanisms or pathways through which spillover effects occur remain unclear. Such health benefits might occur because improvements in the mental and/or behavioral health of one youth might improve overall household dynamics or because improved behavior of one youth might alleviate guilt or stress experienced by caregivers. Alternatively, youth who participate in an evidence-based mental health intervention might also teach skills that they learn to other household members, after which caregivers could potentially improve their own mental health by practicing these skills.

While both qualitative and quantitative approaches can be effectively used to assess the spillover effects of behavioral interventions, qualitative approaches may illuminate many of the unanticipated mechanisms of such effects (Galizzi and Whitmarsh, 2019). Thus, a qualitative approach may be particularly useful in emerging research areas where there is a dearth of pre-existing theory or literature and where a better understanding of potential underlying pathways or mechanisms is needed.

### Current study

This study used a qualitative approach to explore potential mechanisms of mental health spillover effects among primary caregivers of at-risk youth who participated in the YRI delivered within employment programming across rural regions of Sierra Leone. Caregivers of youth participants included partners/spouses, parents, aunts, uncles, and siblings, and all caregivers were interviewed at baseline and follow-up time points. Youth ranged in age from 18 to 30 and demonstrated poor emotion regulation and daily functioning (U19 MH109989). We explored the following questions based on the self-reported perceptions of primary caregivers: (a) whether mental health spillover effects occurred within households of youth who participated in the YRI; (b) potential contributing factors or mechanisms through which mental health spillover effects occurred; and (c) potential sustainment of spillover effects over time.

### Methods

This qualitative study sample includes 20 cohabitating caregivers of youth who participated in the YRI, which was delivered within the context of entrepreneurship training from August through September 2019 (U19 MH109989). All youth who participated in the YRI exhibited difficulties with emotion regulation and daily functioning, as measured by elevated scores on the Difficulties in Emotion Regulation Scale (Gratz and Roemer, 2004) and the World Health Organization Disability Assessment Schedule (WHO, 2014). The YRI is a culturally developed, 12-session evidence-based mental health intervention that incorporates core components of cognitive behavioral and interpersonal therapies to improve emotion regulation, problem-solving and interpersonal skills among youth experiencing compounded adversity (see Betancourt et al., 2014 for more detail). The YRI is delivered to groups of 10–12 same-gendered youth and can be feasibly delivered by lay health workers with quality. The YRI was integrated within an employment promotion program (EPP) created by Deutsche Gesellschaft für Internationale Zusammenarbeit, a German development

agency that has worked with Sierra Leonean youth since 1963 (Betancourt et al., 2021). In the EPP, youth were provided with entrepreneurship and employment skills training and given a stipend to offset costs associated with business start-up. Findings from this larger study are currently under review and indicate youth who participated in the YRI + EPP reported significant improvements in depression and anxiety compared with control youth at post-intervention (Freeman et al., under review). All study procedures were approved by the respective Institutional Review Boards.

### Sampling and recruitment

We recruited youth who had participated in the hybrid-implementation effectiveness trial of the YRI delivered within the EPP (Youth FORWARD/U19 MH109989) to serve as "index participants" ($N$ = 165 YRI + EPP participants; $N$ = 165 control participants) for a closely linked study investigating the indirect benefits of the YRI among peers and cohabiting caregivers (R01 MH117359). Youth "index participants" were recruited across three rural districts in Sierra Leone; Koinadugu, Kailahun and Kono. After providing informed consent, youth "index participants" completed a short survey in which they identified their primary cohabiting caregiver and provided contact information for this individual. Primary cohabiting caregivers were defined as the person whom youth felt emotionally closest to and was primarily responsible for looking after their well-being (Desrosiers et al., 2020), and thus family members could be nominated or other household members, such as a spouse or romantic partner. Nominated caregivers ($N$ = 284) were then recruited and enrolled in the larger study investigating the indirect benefits of the YRI among peers and caregivers (R01 MH117359). To select the sub-sample of caregivers for participation in qualitative interviews in the current study, we used a multivariate sampling matrix in which caregivers were stratified based on age, gender and district of residence ($N$ = 20).

### Eligibility

Inclusion criteria for caregivers were that they should (a) identify as a primary adult caregiver (aged 18 or older) of a Youth FORWARD study participant; and (b) reside in the household of a Youth FORWARD participant. Exclusion criteria were (a) not residing in the household of a YRI participant; (b) severe, active suicidality or psychosis as assessed via the MINI-SCID; and (c) serious cognitive impairments that might inhibit one's ability to comprehend informed consent and participate in the interview. Caregivers who reported active suicidality or symptoms of psychosis were referred for immediate mental health services and followed up by the study social worker. There were no risk of harm cases reported for this sub-sample of caregivers.

### Data collection

Qualitative interviews with caregivers were conducted by trained research assistants using a semi-structured interview guide at two different time points. Interviews were conducted in Krio, the predominant local language, or in the local dialect that caregivers felt most comfortable speaking in if they were not fluent in Krio. Interviews focused on exploring (a) whether mental health spillover effects had occurred following youth participation in the YRI; (b) potential mechanisms through which spillover effects may have

occurred (i.e., reduced sense of caregiving burden and improved household dynamics); and (c) potential sustainment of spillover effects over time. The semi-structured guide included questions such as the following:

1) *How did having (participant's name) in the program affect how you felt?*
2) *Have you noticed any changes in how you feel since (participant's name) completed the YRI program? If so, what changes have you noticed? Why do you think you feel differently?*
3) *What, if any, change did you observe from (participant's name) participation in the YRI?*
4) *How did having (participant's name) in the program affect the household relationships between everyone who lives here? Have you noticed changes in the household? If so, what is different? How was it before?*

All interviews were audio-recorded. The first interview was completed at the post-YRI assessment timepoint (all youth participants had completed the YRI), between September 2019 and January 2020. A follow-up interview was completed between December 2020 and January 2021. The purpose of the two time points was to explore whether any changes had occurred over time in terms of reported spillover effects. In addition, follow-up interviews allowed the research team to assess if reported changes after the YRI had been sustained over time. Interviews ranged from 30 to 60 min in duration, and caregivers were compensated for their time via a household gift (cooking oil, soap, rice etc.) worth SLL 30,000.

### Data analysis

Forty interview transcripts were first transcribed in Sierra Leonean Krio (or Mende) and translated to English with all identifying information removed. Translations were cross-checked by another native Sierra Leonean Krio speaker. A combination of grounded theory along with thematic content analysis was used for data analysis (Strauss, 1987; Anderson, 2007). In this approach, themes were derived from the data itself but guided by the research questions. Before beginning the data analysis process, two research team members read each transcript in depth and used an "open coding" method to write memos and notes on the themes and patterns emerging from the data. Next, team members discussed the themes and placed them into categories according to the research questions. Categories were encompassed in a three-level codebook, guided by the Boyatzis' approach (Boyatzis, 1998), in which there are levels of codes, definitions and examples for each code, and inclusion and exclusion criteria for each code. The codebook was developed through an iterative process that required team members to develop and test several versions of the codebook on a subset of transcripts until the codebook was finalized. Once the codebook was finalized, inter-coder reliability was tested between the two coders on a subset of transcripts ($n$ = 3) (MacPhail et al., 2016; O'Connor and Joffe, 2020). After satisfactory inter-coder reliability was reached, team members finished coding transcripts separately. All coding was completed in MaxQDA (VERBI Software, 2021). Throughout the coding process, before and after establishing inter-coder reliability, team members met weekly to discuss transcript memos and any difficulties with thematic content analysis in sections of transcripts. After coding was completed, team members used axial coding to identify key relationships between and within themes.

**Table 1.** Demographic data for caregivers and youth participants

| Caregiver relationship | Individual | Gender | Age | District | # in HH | Employment | Highest level of education |
|---|---|---|---|---|---|---|---|
| Uncle | Caregiver | M | 45 | Koinadugu | 8 | District office worker | None |
| | YRI participant | F | 23 | Koinadugu | 3 | – | Senior secondary |
| Mother | Caregiver | F | 58 | Koinadugu | 7 | Farmer | – |
| | YRI participant | M | 30 | Koinadugu | 9 | Farmer | Some senior secondary |
| Husband | Caregiver | M | 31 | Kono | 4 | Farmer | Some junior secondary |
| | YRI participant | F | 24 | Kono | 6 | Petty trading | Some junior secondary |
| Friend | Caregiver | F | 42 | Kono | 3 | Farmer | – |
| | YRI participant | M | 18 | Kono | 10 | Farmer | Some primary |
| Mother | Caregiver | F | 45 | Kono | 7 | Farmer | – |
| | YRI participant | M | 24 | Kono | 6 | Farmer | Junior secondary |
| Husband | Caregiver | M | 34 | Kono | 7 | Farmer | – |
| | YRI participant | F | 29 | Kono | 5 | Petty trading | None |
| Wife | Caregiver | F | 25 | Kailahun | 10 | Farmer | Primary |
| | YRI participant | M | 30 | Kailahun | 14 | Farmer | None |
| Aunt | Caregiver | F | 24 | Koinadugu | 9 | Farmer | Some primary |
| | YRI participant | F | 25 | Koinadugu | 8 | Petty trading | Some junior secondary |
| Wife | Caregiver | F | 22 | Koinadugu | 3 | Farmer | Some junior secondary |
| | YRI participant | M | 25 | Koinadugu | 3 | Petty trading | Senior secondary |
| Mother | Caregiver | F | 41 | Kono | 14 | Soldier | Primary |
| | YRI participant | F | 26 | Kono | 13 | Miner | Some junior secondary |
| Brother | Caregiver | M | 46 | Kono | 4 | Farmer | – |
| | YRI participant | M | 29 | Kono | 5 | Farmer | None |
| Mother | Caregiver | F | 50 | Koinadugu | 4 | Farmer | – |
| | YRI participant | M | 27 | Koinadugu | 6 | Carpenter | None |
| Father | Caregiver | M | 53 | Kailahun | 8 | Tailor | – |
| | YRI participant | M | 20 | Kailahun | 4 | Farmer | Some senior secondary |
| Wife | Caregiver | F | 34 | Koinadugu | 7 | Farmer | – |
| | YRI participant | M | 26 | Koinadugu | 7 | Mechanic | Some junior secondary |
| Father | Caregiver | M | 35 | Koinadugu | 9 | Security | – |
| | YRI participant | F | 19 | Koinadugu | 5 | Petty trading | None |
| Husband | Caregiver | M | 42 | Koinadugu | 8 | Farmer | – |
| | YRI participant | F | 20 | Koinadugu | 6 | Farmer | None |
| Wife | Caregiver | F | 27 | Kono | 6 | Farmer | – |
| | YRI participant | M | 25 | Kono | 12 | Farmer | Some junior secondary |
| Uncle | Caregiver | M | 41 | Kono | 6 | Police inspector | Some primary |
| | YRI participant | F | 21 | Kono | 4 | – | Some junior secondary |
| Mother | Caregiver | F | 44 | Kono | 7 | Miner | – |
| | YRI participant | M | 20 | Kono | 10 | – | Senior secondary |
| Mother | Caregiver | F | 50 | Kailahun | 6 | Farmer | – |
| | YRI participant | F | 20 | Kailahun | 5 | Petty trading | Primary |

Abbreviation: YRI, Youth Readiness Intervention.

## Results

Nominated caregivers of YRI participants were related to participants in a variety of ways, including mothers ($N = 6$), fathers, ($N = 2$) spouses/intimate partners ($N = 7$), aunts/uncles ($N = 3$), an older friend ($N = 1$) or a sibling ($N = 1$). Ages of YRI participants ranged from 18 to 30, and ages of caregivers ranged from 24 to 58. Individuals were sampled from Kailahun, Kono and Koinadugu districts of Sierra Leone, and each YRI participant lived in the same district as their nominated caregiver. The average household size was seven members per household. Most caregivers and YRI participants identified farming as their primary income-generating activity, though other income-generating activities included petty trading and mining. Very few caregivers or YRI participants had finished senior secondary school, and many reported no formal education at all. All identified caregivers participated in both the post-YRI and the follow-up interviews. See Table 1 for full demographic details of primary cohabitating caregiver (e.g., mother, father, spouse, sibling, aunt, and uncle) and youth participant dyads in the study.

Results of qualitative analysis suggested that mental health spillover effects likely occurred among YRI participants' caregivers, based on their self-reported perceptions. Results also supported three potential mechanisms through which spillover effects occurred: reduced burden of care, behavior changes in YRI participants and improved interpersonal relationships. Figure 1 represents a model of the spillover effects experienced by caregivers.

### Mental health spillover effects

#### Improved sense of caregiver well-being

Based on the caregiver reports, a reduced sense of burden, changes in YRI participant behavior and improved interpersonal relationships contributed to improvements in caregivers' emotional well-being over time. Caregivers frequently used the term "happiness" to describe their improved sense of well-being. When asked about their interaction with YRI participants before the intervention, most caregivers, regardless of the type of caregiving relationship, discussed feeling stressed due to difficulties with youth behavior and mood. For example, *"A parent would not want to hear people saying every day, 'Look your child has caused trouble, your child has done that', a parent does not want that"* (father of YRI participant).

Similarly, an intimate partner stated, *"You know that, whenever there is quarrel you will not feel good…My heart was pounding and when I sit, I was thinking about it. That was how it happened to me"* (husband of YRI participant).

When caregivers described changes in the YRI participant's behaviors, they described the changes in their emotional well-being and happiness as directly linked to these changes in youth behavior:

*So, my happiness is because of his concentration now, he has partaken in this program. And I'm praying that they concentrate, let his heart be in it and let him do it better than (before)* (father of YRI participant).

*As she has participated in the program… Now I am feeling good, and as I sit, I smile about it. Now we sit and have fun together. When people are seeing us together, they are saying husband and wife are having fun and there is time for everything* (husband of YRI participant).

### Mechanisms of spillover effects

#### Reduced burden of care

Caregivers spoke about the YRI participants in relation to their caregiving burden before and after youth participation in the YRI. During follow-up interviews, caregivers provided similar examples of reduced burden of care and stated that these effects had continued between interview timepoints. Caregivers described how the YRI participant's behavior had caused a sense of burden prior to their participation in the YRI and EPP, which was largely related to poor engagement in school and work: *"He was a man I sent to school, he didn't go, I sent him to the Arabic school, he didn't go"* (father of YRI participant) and *"Previously, when I would tell him, like, '(Index Participant) you should go to the farm', he would ask, 'Who?' and he would not go"* (mother of YRI participant). After YRI participation, caregivers described a sense of "happiness" and an alleviated sense of burden because the youth increased their engagement in household responsibilities (i.e., caring for other members of the household) and attending school and work. One caregiver, YRI participant's uncle, described this at timepoint 1:

*It has really made me feel good, because I'm no longer doing what I used to do in my house – waking up the children, to tell them what next… I think I'm now resting; she is the one that has taken up such responsibility.*

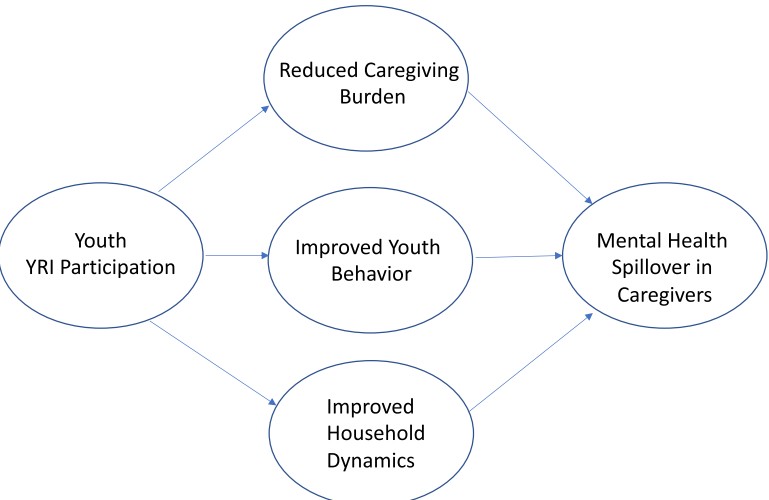

**Figure 1.** Mechanisms contributing to mental health spillover effects among caregivers.

Eight months later in the follow-up interview, this uncle stated,

*She is helping me at the house…. She understands me now, she knows when I am broke and so she will step in and prepare food. Before the YRI, no – you would not even see her close to the kitchen. She didn't like cooking or laundry, but now she does. These are the changes I have experienced with her.*

### Behavior changes from YRI participants

When asked about youth behaviors before and after participating in the YRI, caregivers gravitated toward speaking about the youth participant's mood, drinking and smoking habits, disobedience, and engagement in education. Caregivers provided examples of perceived improvements in each of these behaviors, which were sustained at follow-up. Many caregivers described the youth participant before the YRI as temperamental and quick to quarrel, "*She used to be very hot tempered*" (mother of YRI participant) and "*… she was not a kind person, she does not care about people… and she will also become angry when you bring up issues to her*" (aunt of YRI participant). Caregivers of male YRI participants spoke about youth drinking alcohol and smoking marijuana. These behaviors were often described with a negative connotation and attributed to the YRI participants' overall lack of engagement;

*Well at the time he was drinking, he would go and misbehave with his colleagues and bring the problem to me. Or when he smokes, he would go and misbehave with his colleagues and then bring that problem to me* (brother of YRI participant).

Caregivers described YRI participants as uncooperative and disrespectful before attending the YRI. One caregiver, a YRI participant's mother, stated, "*Previously… he was not listening to my advice, and wouldn't do anything I do him to do for me. He had no respect for me, and was confrontational…but he has now changed.*" YRI participants were described as behaving better and being more respectful at both timepoints. This caregiver also referenced the sustained behavior changes observed among youth and stated in the follow-up interview, "*As I explained during the last time you came, (YRI participant) has changed from the way he used to behave. He still heeds my advice.*"

Caregivers mentioned a lack of engagement in the household among youth and said that they were often "going out" of the house before participating in the YRI. This was also described as the YRI participant being "wayward" during the post-YRI interview: "*…at first she was wayward and that was the lifestyle. Roaming about. I see now she is not roaming about anymore. When she comes back, she stays at home*" (mother of YRI participant). At follow-up, the same caregiver stated, "*since she went to the training, she has not been roaming about anymore.*"

Caregivers expressed concern over the YRI participants' lack of involvement in formal and informal educational opportunities prior to YRI participation and discussed how youth were more engaged in school after completing the YRI (at both post-YRI and follow-up timepoints). For example, one caregiver stated, "*When I sent him to school, he did not go further with it, he has no skill*" (father of YRI participant). Similarly, a YRI participant's mother stated, "*…Before it was not easy to take her book and study, but now I do see her reading her book sometimes at night.*"

### Improved household dynamics/interpersonal relationships

All caregivers described improvements in their relationships with YRI participants. Caregivers provided examples of increases in comradery, more effective communication with less quarreling and mutual encouragement during both interview timepoints.

Additionally, caregivers explained positive changes in how YRI participants related with other adults and siblings in the household, decreased conflict or behavioral issues within the greater community and shifts in peer relationships that were perceived as positive.

Six caregivers described situations in which YRI participants demonstrated improved interpersonal functioning that contributed to increased experiences of positive emotions and a sense of well-being among caregivers. For example, one husband of a participant explained his relationship with his wife post-YRI, "*Now she encourages me, and it (is) very difficult for us to get into (a) quarrel.*" Eight months later, he described their relationship again as follows: "*she has stopped quarreling and she talks nicely to me. Her friends have also started spending more time with her. I visit their husbands and we all laugh and play together.*"

Another caregiver provided an example of his niece offering emotional support after her time in the YRI:

*She would even come to me and ask, "Uncle, what happened, is it any problem at your work?" But I would tell her no, and she would say, "It's because I see that you are sad…" and she would joke with me so that I would feel happy* (uncle of YRI participant).

Six caregivers mentioned scenarios in which YRI participants demonstrated more thoughtfulness and respect in their relationships with caregivers, which contributed to reduced hostility and increased positive emotions. For example, one YRI participant's wife described her relationship with her husband after his participation in the YRI:

*He and I were not united until this program came and he started attending… on the other occasion, we nearly quarreled as he was about to go for the program. They started the training…when he came from there, he started talking to me nicely; the other time, he even apologized to me.*" She continues to say "*At first, we (didn't) relate well, but now, we sit together and chat, play and laugh. He no longer beats me. Have you seen?*" (wife of YRI participant).

Nine caregivers provided examples of their households becoming more peaceful and less stressful as YRI participants began helping other household adults with household chores, or speaking more respectfully. Caregivers used the phrase "cordial" and "peaceful" to describe the household environment as a result of the YRI participant's behavior. One caregiver described how the YRI participant improved his relationships with household members:

*When (he) came from the program… I have said it, not only me and my children have seen the changes in him, everyone at home has seen it, even his tenants. He takes care of my relatives, there is no problem between them though it was not like that. Before he participated in this training, my relatives were afraid of him and he was not speaking with them but now… the girl who was standing here is my younger sister. Now they play and laugh* (uncle of YRI participant, age 41). In the follow-up interview, he told the interviewer, "*Since the last time you came here, I still notice the same changes.*"

Finally, caregivers offered examples of positive changes between YRI participants and others in the community or in their peer groups. Six caregivers reflected on comments from community members regarding the YRI participants' behavior in the community. For example, one YRI participant's mother stated that neighbors had complained about the YRI participant prior to their participation in the intervention, but now they no longer complained. Another caregiver recalled how the YRI participant had been unreliable at their workplace and how they used to hear gossip around town about this person's behavior, but "*now all of that has changed*" after the YRI (wife of YRI participant).

Regarding peer relationships, many caregivers expressed gratitude and relief that youth were no longer hanging around with peers perceived as bad influences, which contributed to caregivers experiencing less stress. For example, one caregiver stated, "*when we say, don't go (with bad friends), he won't go there; he now heeds our advice. I am really happy about that*" (mother of YRI participant).

## Discussion

This study used a qualitative approach to better understand potential mechanisms of mental health spillover effects among cohabitating partners and caregivers of youth who participated in an evidence-based mental health intervention in rural regions of Sierra Leone. Findings show that caregivers and partners experienced an improved sense of general well-being related to improvements that they observed in youth behavior and functioning following YRI participation. Although results should be interpreted with caution given the small sample size, findings from the eight-month follow-up interviews suggest that mental health spillover effects appeared to be sustained over time. The current study fills gaps in prior quantitative studies investigating mental health spillover effects by identifying several specific mechanisms through which the reported mental health spillover effects occurred among cohabitating caregivers and partners.

Caregivers attributed their reduced sense of strain and burden to behavioral changes in the youth participant, reduced burden of care and improved relationships with YRI participants. These findings are consistent with prior quantitative research indicating that caregivers of youth participating in the YRI in Sierra Leone reported significant reductions in the sense of caregiving burden post-intervention compared with caregivers of control youth (McBain et al., 2015). Qualitative findings deepen prior research by highlighting improvements in youth behavior as a potential mechanism through which caregivers and partners experience a reduced sense of caregiving burden, and reduced sense of burden as a potential mechanism driving overall improvements in caregiver/partner well-being.

Caregivers typically associated reduced burden of care with tangible changes in the ways that YRI participants assisted with household chores or income-generating activities. Other studies in LMICs which examine the connection of mental health with economic and instrumental support in the home are broadly consistent with these findings (Naidoo et al., 2019; Knapp and Wong, 2020; Zimmerman et al., 2021).

As caregivers and partners reported improvements in relationships with YRI participants, they also discussed positive changes in overall household dynamics as a function of YRI participation. Family and household dynamics are generally associated with improved mental health outcomes for caregivers (Puffer et al., 2020). Improved household dynamics are notable given that the average household size was seven people. In the rural regions of Sierra Leone where this study was conducted, houses are typically small and crowded, with 55% of households consisting of one or two rooms and most including young children (Macarthy et al., 2017). Given that caregivers and partners did not explicitly report that youth participants shared skills or techniques that they learned from the YRI with family members, findings suggest that improving the functioning of one household member struggling with emotional/behavioral difficulties or mental health problems may have a much wider benefit via spillover effects. Future research should better explore the wider benefits of evidence-based mental health interventions for other household members occurring via spillover as well as through which mechanisms these occur. Because Sierra Leone is a collectivist culture (Hofstede, 2020), where the concept of family often extends far beyond the nuclear family, including non-biological relations, the potential to leverage the phenomenon of mental health spillover effects to expand the reach of evidence-based mental health interventions across a wider segment of society may be greater there, as well as in other collectivist cultures throughout the world.

Extant data demonstrates that mental health and behavior-change outcomes post-interventions in LMICs can be sustained over the long term (Nadkarni et al., 2017; Weobong et al., 2017; Pilling et al., 2020). Future research is needed to better unpack the mechanisms that might facilitate and support the sustainment of mental health benefits over time, including those occurring indirectly through spillover effects, which can in turn extend the reach of evidence-based interventions to a larger proportion of the population.

## Limitations

This study has several limitations to note. First, the relatively small sample size of participants who resided in a very rural region of Sierra Leone limits the generalizability of study finds to other populations in LMICs or other resource-constrained settings, such as those residing in more urban areas, those with higher levels of education or household income, or those with greater access to technology and network connectivity. Finally, it is important to acknowledge that the YRI was delivered as a component of an entrepreneurship program. Our qualitative interviewers were trained to focus the interviews specifically on YRI participation. The interview guide contained a preamble that clearly described that the focus of the interview was to understand the spillover effects of the YRI and *not* the employment programming. If caregivers shared observations that appeared to reflect participation in the employment programming (EPP) component, interviewers asked the caregiver to clarify which program they were referring to. Data analysts coded any references to the EPP so that data linked to these codes could be excluded from the current study. Despite these efforts, it is possible that some caregivers were unable to distinguish which program component facilitated the observed changes in YRI participants' behavior and therefore some results may be related to EPP participation and not be solely attributed to YRI participation. Future analyses on mental health spillover effects in LMICs would benefit from including additional household members in qualitative data collection and also from better exploring the potential influence of cultural factors on the phenomenon of mental health spillover effects. For example, stigma regarding mental illness in Sierra Leone (Akinsulure-Smith and Conteh, 2018) could impact the extent to which caregivers may openly discuss their feelings of anxiety, stress or depression related to caregiving burdens.

Despite these limitations, our study also includes several strengths, such as our longitudinal study design with two data collection points and our retention rate (19/20 interviews completed at follow-up). The longitudinal study design allowed us to understand caregiver beliefs and attitudes after the intervention was delivered, when changes in YRI participants may have been most noticeable, as well as how mental health spillover effects might be diminished or sustained over time. Further, nominated caregivers represented a variety of relationship types, ranging from spouse/

partner, to mother to sibling, thus expanding the conceptualization of who might experience mental health spillover effects in a low-resource, rural region of Sierra Leone.

## Conclusions

Based on self-reports of primary cohabitating caregivers of youth who participated in an evidence-based mental health intervention, (a) caregivers experienced mental health spillover effects following youth participation in the evidence-based mental health intervention linked with employment programming; and (b) mental health spillover effects appeared to be sustained over time. In a setting like Sierra Leone where mental health issues are highly stigmatized, attending to and utilizing such spillover effects can provide bonus mental health support without the associated stigma. The mechanisms of spillover effects occurring in the YRI and similar evidence-based mental health interventions may help address the challenge of delivering innovative interventions with sufficient breadth and depth to benefit large populations or large geographic areas in LMICs and other low-resource settings. Evidence for mental health spillover effects among caregivers and partners may spur further research quantifying the indirect benefits of mental health interventions among household members and across wider social networks, including factoring mental health spillover effects into future cost-effectiveness analysis. This information could serve to influence policymakers' decisions to increase investments in mental health services and the infrastructure to support them in LMICs.

**Open peer review.** To view the open peer review materials for this article, please visit http://doi.org/10.1017/gmh.2023.36.

**Data availability statement.** The datasets used and analyzed in the current study can be made available upon reasonable request to the Principal Investigator.

**Author contribution.** All authors made substantial contributions to (a) the conception or design of the work; and/or (b) acquisition of the data; and/or (c) analysis and interpretation of the data for the manuscript. All authors contributed to drafting and/or revising the manuscript critically for important intellectual content. All authors provided final approval of the version to be published and agree to be accountable for all aspects of the work.

**Financial support.** This study was funded by the National Institute of Mental Health (Grant No. R01 MH117359).

**Competing interest.** The authors declare that they have no competing interest.

**Ethics standard.** This study received ethical approval from the Boston College Institutional Review Board and the Sierra Leone Ethics and Scientific Review Committee.

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
