## [Reviewer Report]

January 10, 2023

Dear Editorial Board,

I would like to submit the attached manuscript, entitled “Exploring Potential Mental Health Spillover Effects among Caregivers and Partners of Youth in Sierra Leone: A Qualitative Study”, for consideration in Global Mental Health. 

This manuscript reports findings from a qualitative study exploring the phenomenon of mental health spillover effects among primary cohabitating caregivers of high-risk youth who participated in an evidence-based mental health intervention in Sierra Leone. Findings have implications for policy makers and other key stakeholders in terms of investment in evidence-based mental health services in LMICs that may offer wider societal benefits that extend beyond those directly receiving services. This study has received Institutional Review Board approval. 

The content of this manuscript has not been published or submitted for publication elsewhere in the same form in any language beyond this submission to Global Mental Health. All authors have approved the manuscript for submission, there are no competing interests, conflicts of interest, or issues relating to journal policies. 

I will be serving as the corresponding author for this manuscript, and I will inform the coauthors about the status of the manuscript, including editorial decisions and content of revisions. Thank you for considering this manuscript for publication in Global Mental Health. We look forward to hearing from you. 

Kind Regards,

Alethea Desrosiers, Ph.D.

Assistant Professor

Brown University

Warren Alpert Medical School

Providence, RI 02906

alethea_desrosiers@brown.edu

---

## [Reviewer Report]

This paper reports on a small interview study conducted in conjunction with a larger study about which we learn little; both studies, it is implied, look at mental health spillover effects from mental health interventions with youth. More information is needed to fully understand what has been done. The questions touched on in the abstract appear to be ones that would best be answered with a quantitative approach. It is not clear what the mental health component of the intervention received by young people was, and whether it was found to be effective, other than in these 20 interviews. The review refers to ‘evidence-based intervention’ but that evidence doesn’t seem to be explained. The intervention seems to be related to employment. What where the findings of the main study? What were the findings of the quantitative analysis of spillover effects? Were benefits found to participants and caregivers in the larger study? Once this is established, the authors should then consider what a qualitative approach can add to what has been investigated in the quantitative work.

The authors have interview data from 20 caregivers in Sierra Leone, which could potentially provide valuable insights. What is more there is a longitudinal aspect to the study design with participants being followed up a second time. It would be good to see that longitudinal aspect reflected in the research questions and incorporated more into the analysis. Were the youth also followed up? Is this reported anywhere? What were the findings for the youth involved?

The information on how the sample was arrived at is limited and raises many questions. How did youth come to be involved in the original study? Although it’s not commented on it seems you had 100% follow up rate. Is that correct? Or were only those with 2 interviews chosen post hoc to form the sample? There is also no mention of anyone not benefitting from the interview – were only participants who benefitted from the intervention included?

The themes are no doubt important, but could all perhaps be attributed to the passing of time and growing maturity of the young people. The qualitative data might provide more opportunity to unpick this question, if the quantitative study did not have a control group – it would be good to see this reported. To understand what is going on here, the reader needs to know more about the programs involved.

The discussion section again starts off sounding like a quantitative study, all participants experienced spillover effects, all were sustained at 8 months etc. The discussion could explain better what this study adds over the 2015 study.

The discussion suggests the findings could support the addition of a booster session for the young people, but I did not see anything presented that backs this up, especially since we do not know anything about any mental health problems the young people had or the benefits of the programme to them. Likewise, there are big claims in relation to the specific importance of spillover effects to Sierra Leone because they have a ‘collectivist culture’ compared to other LMICs. This would have been very interesting to investigate and if the authors have data to investigate this, it would be great to include this in the paper – at present nothing in the results presented appears to relate to collectivism and there is not even a reference offered to support the suggestion the country is more collectivist than others.

Reference is made to quant findings from other studies, but why not from the study of which this was part? The fact that the limitations refer to difficulties tracking and finding people rather indicates we are not told the full story of how the sample with 100% follow-up was arrived at.

What does the sentence re ‘redirect interviews back to the YRI’ mean? Are we to understand that some may not have received the program at all? What is the relevance of the YRI? There does not seem any material presented which attempts to relate benefits to specific YRI components. Given this, ‘Some results may not be solely attributed to YRI’ participation' seems like an understatement.

The authors refer to the non-generalisability of findings due to the specific location in rural Sierra Leone and yet the findings themselves seem very generic and could apply anywhere around the world one would think. It would be fascinating to have heard something about the specific cultural context and how this affected ‘spillover effects’. The authors evidently feel this to be the case, but have not yet explored these interesting issues in this paper.

---

## [Reviewer Report]

This paper explores the spillover effect of an intervention in rural Sierra Leone specifically focused on the cohabitating caregivers of those who received the intervention. This is an important issue as the field explores the potential of reaching families and communities. The results are also interesting and informative.

The primary area for improvement of this paper is in its need for more clarity from the beginning about the research question, scope, and sample.

Abstract

The abstract is a bit unclear in terms of who the intervention participants are. It can be interpreted that one caregiver participated – and that the spillover is related to the other caregiver (cohabitating). I realized this was not the case after reading the Current Study section.

Two other minor suggestions to consider: (1) The first statement focuses on “wider societal benefits,” which is true, but this study focuses on the value of within-household spillover, which is important in and of itself. At first, I expected the study to talk about more community-level spillover given this opening. (2) If it is possible within the space, having a brief mention of the mechanisms through which participants described their change would then include the unique aspect of this study in looking at the process of spillover. The opening of the Discussion provides a sentence that may be useful to adapt for the Abstract.

Introduction

1. Authors should provide a citation that clearly supports the 98% treatment gap.

2. Related to the confusion in the abstract, it would be helpful to make the Current Study section clearer in terms of who participated directly in the intervention and who participated in this follow up study. For example, the idea of “partners” is not included but it is not clear whose partners (older youth vs. partners of the caregivers of the youth who participated). One idea for consideration is perhaps to avoid using “cohabitating” since that is often to refer to adult-adult relationships.

3. Authors should consider including a hypothesis—or, if framed as exploratory, some possible ideas for mechanisms through which spillover could occur. This would solidify the rationale for the research question and help the reader understand the types of associations you are examining.

Methods:

4. Minor: Typo on page 6 (spend verses sense).

5. The sample remained unclear even after the Methods. It was in Results that it became clear that some of the youth had partners/spouses who were participating. One reason this was somewhat surprising was also that “caregiving burden” had been mentioned several times. If that is “caregiving” because of living with someone struggling with mental health problems (rather than general caregiving in terms of parent-type roles), that should also be defined earlier. (Defining “youth” earlier in terms of extending to age 30 may also be helpful in providing the reader with key background on the sample.)

6. The description of the interview, including sample questions, includes a description of other goals besides examining mental health spillover (ie, also mentions household dynamics and youth behavior), without identifying these are potential mechanisms (which becomes clear later in Results and Discussion). It would be helpful to know which questions/topics were used in the current analysis and why. For example, were questions about changes in the youth asked in order to explore whether these influenced caregiver mental health? Or perhaps these were the qualitative questions asked for many purposes, and then this spillover emerged in an unexpected way? It is also curious that there were not more direct questions about spillover in terms of whether the intervention youth participant talked about the intervention content at home.

7. Data Analysis: Authors state that the intervention’s theory of change influenced the analysis, and it would be helpful to know more about that. Often the theory of change has the endpoint of the outcomes at the participant-level; was this then extended in a way that related to spillover, or was the theory of change used for the part of the qualitative study just looking at youth outcomes? (This goes back to the questions in the comment just above about which questions and data fed into the specific analysis/results for the research question addressed in this analysis vs. in the qualitative component of the study overall.)

8. Data Analysis: It would be helpful to know how inter-coder reliability was addressed. It states that several versions were tested on a subset of transcripts, but it is unclear what that would mean and whether it would adequately support the decisions that coders could then code separately in a way that was consistent.

Results

9. Minor: It is stated that household sizes vary widely. Having data on the range here would be relevant since the overall topic is spillover/reach. In Discussion, I believe a mean of 7 is mentioned, which could be stated here.

10. Overall, the use of the pseudonyms (especially multiple in one quote) seems unnecessary. Authors should consider instead focusing on participant and “caregiver” characteristics/relationships. This is especially important given that youths’ spouses are included, meaning that “caregiving” would mean something quite different. For example, it would be helpful to say things like, “The husband of a 25-year-old participant said…” (no names needed since there’s no narrative presented about specific people).

11. Major: It is unclear how the second theme of the behavior change among the youth themselves is related to mental health spillover effects on caregivers. Caregivers are of course glad to observe these improvements, but are reporting on the youth rather than the mental health benefits they experience due to this improvement. Later, it becomes clear in the Discussion that this is seen as a mechanism, but within Results, it reads just as qualitative results supporting intervention effects on the youth as reported by caregivers.

12. The same is true to some extent with the Interpersonal Relationships theme, though it is more clear how the caregivers are enjoying these changes as recipients of these new interpersonal behaviors. It is unclear whether this would be “mental health” spillover effects, though, as participants are not describing how these changes make them feel.

13. In contrast, the Caregiver Wellbeing is directly related to the stated research question of mental health spillover, and what is expected based on the framing of this manuscript. One way to partially address the confusion about aims/scope might be to present the mental health benefits first (ie, caregivers increased happiness) and then present the other themes (improved behavior, better relationships) as contributors (mechanisms) to this improved wellbeing of caregivers—and to frame the paper to identify the mental health spillover and the factors that contribute to that.

Discussion

14. The first two sentences are very clear and could be useful to inform framing the aims, as well as providing a brief statement of mechanisms-related findings in the Abstract.

15. The point about participants not describing that the youth passed along lessons/skills learned in the intervention is important and gets a bit buried within the Discussion. It would be helpful in the Introduction to know whether the researchers expected that to be one pathway in spillover (e.g., youth teach coping skills to their parents/spouses), as that would be a very rationale expectation and the typical understanding of what “spillover” means. It is quite an interesting finding that this did not emerge—that the mental health benefits came about indirectly as related to the improvements in the direct participant.

16. One limitation to mention, related to the above, is that the interviews do not seem to have covered questions directly related to whether the youth participants did try to teach things that they learned—or whether the caregivers had at least discussed what the youth learned. If these questions had been asked, perhaps some of those expected spillover pathways might have been seen.

---

## [Reviewer Report]

The authors have addressed the reviewers' concerns and the paper is improved. In my view it is suitable for publication. The authors may like to consider further moderating some language.

In the abstract I am not sure it is quite correct to say the study explores ‘ whether’ spillover effects had occurred. You could say whether they were perceived to have occurred? Or if more appropriate to what was actually done, maybe whether there ‘appeared to have been’ any spillover effects'?

Equally, rather than whether effects are sustained over time, you may have been exploring whether ‘perceived’ changes appear to be maintained over time.

Perhaps authors do not need to assert, now more strongly, which programme was the cause of any perceived improvements. They can’t perhaps really know, and just because the other programme did not contain specific mental health components, does not mean it could not have contributed to improved mental health.

---

## [Reviewer Report]

Please would you address the additional comments by the reviewer before the manuscript can be accepted. In addition, please would you review your explanation of inter-rater reliability. The second reviewer had asked about inter-coder reliability. Please explain what you did and provide a reference supporting your approach.

---

## [Reviewer Report]

June 14, 2023

Dear Dr. Petersen and the Referees,

Thank you very much for the invitation to revise and resubmit our manuscript, entitled “Exploring Potential Mental Health Spillover Effects among Caregivers and Partners of Youth in Sierra Leone: A Qualitative Study”, for consideration in Global Mental Health. We appreciate the comments from the referee(s) and have responded in detail below. Revisions to the manuscript in response to referee comments appear in track changes. Please do not hesitate to contact me if there are any questions or concerns. 

I will be serving as the corresponding author for this manuscript, and I will inform the coauthors about the status of the manuscript, including editorial decisions and content of revisions. We look forward to hearing from you. 

Kind Regards,

Alethea Desrosiers, Ph.D.

Assistant Professor

Brown University

Warren Alpert Medical School

Providence, RI 02906

alethea_desrosiers@brown.edu

Comments:

1. Please would you review your explanation of inter-rater reliability. The second reviewer had asked about inter-coder reliability. Please explain what you did and provide a reference supporting your approach.

We thank the reviewers for bringing this to our attention. After further review, we realized that we used the term inter-rater reliability incorrectly, which is commonly conflated with inter-coder agreement (O’Conner & Joffe, 2020). We tested inter-coder agreement, not inter-rater reliability, during qualitative analysis. We have corrected this in our manuscript. We have added references to our manuscript that support the approach that we took regarding inter-coder reliability (MacPhail et al., 2016; O’Conner & Joffe, 2020). The references we have sited also describe the steps for developing a codebook and meeting regularly as a team to discuss themes and how codes are being applied to represent themes. 

2. In the abstract, I am not sure it is quite correct to say the study explores ‘ whether’ spillover effects had occurred. You could say whether they were perceived to have occurred? Or if more appropriate to what was actually done, maybe whether there ‘appeared to have been’ any spillover effects'?

As per the reviewer’s suggestion, we have adjusted our phrasing in the abstract to be less assertive. We have changed the phrase: ‘whether spillover effects had occurred’ to read:

“Qualitative interviews were conducted at two time points to explore: (a) potential mental health spillover effects and (b) through which mechanisms spillover may have occurred.”

This is more consistent with our phrasing of exploring potential mental health spillover effects earlier in the abstract.

3. Equally, rather than whether effects are sustained over time, you may have been exploring whether ‘perceived’ changes appear to be maintained over time.

We have adjusted our language regarding effects over time as per the reviewer’s suggestion. We added the following phrase to make it clear that exploration of our research questions was based on the self-reported perceptions of caregivers. We added the following phrase prior to the statements about ‘whether effects were sustained over time’ in the introduction

“We explored the following questions based on the self-reported perceptions of primary caregivers”. 

We also changed the phrasing in the first paragraph of the conclusion regarding sustainment of mental health spillover effects over time to read:

“…appeared to be sustained over time”.

We believe that these modifications make it clearer that findings are based on perceptions of caregivers and their self-reported experiences.

4. Perhaps authors do not need to assert, now more strongly, which program was the cause of any perceived improvements. They can’t perhaps really know, and just because the other program did not contain specific mental health components, does not mean it could not have contributed to improved mental health.

In the limitations section, we note the possibility that effects could have occurred due to participation in the employment promotion program. We agree that we cannot know for certain that the other program did not contribute to mental health improvements.

“…it is possible that some caregivers were unable to distinguish which program component facilitated the observed changes in YRI participants’ behavior and therefore some results may be related to EPP participation and not be solely attributed to YRI participation.”

We also modified our language in the conclusion section to note that the evidence-based mental health intervention was linked with employment programming.